# Metabolomic Characterization and Bioinformatic Studies of Bioactive Compounds in Two Varieties of *Psidium guajava* L. Leaf by GC–MS Analysis

**DOI:** 10.3390/ijms26062530

**Published:** 2025-03-12

**Authors:** Ana Victoria Valdivia-Padilla, Ashutosh Sharma, Jorge A. Zegbe, José Francisco Morales-Domínguez

**Affiliations:** 1Centro de Ciencias Básicas, Departamento de Química, Universidad Autónoma de Aguascalientes, Aguascalientes 20131, Mexico; al97640@edu.uaa.mx; 2Centre of Bioengineering, NatProLab, School of Engineering and Sciences, Tecnologico de Monterrey, Querétaro 76130, Mexico; asharma@tec.mx; 3Campo Experimental Pabellón, Instituto Nacional de Investigaciones Forestales, Agrícolas y Pecuarias, Km 32.5 Carretera Aguascalientes-Zacatecas, Pabellón de Arteaga, Aguascalientes 20668, Mexico; zegbe.jorge@inifap.gob.mx

**Keywords:** medicinal plants, *Psidium guajava* L., bioactive compounds, guava leaves, molecular docking

## Abstract

The guava tree (*Psidium guajava* L.) is a tropical plant from the Myrtaceae family. Leaf extracts from this plant have been used in traditional medicine to treat gastrointestinal disorders and exhibit several functional activities that benefit human health. Different varieties of guava trees produce fruits in colors ranging from white to red and present a characteristic metabolic profile in both their leaves and fruits. This study presents a metabolomic characterization of the leaves from two guava varieties: the Caxcana cultivar with yellow fruits and the S-56 accession with pink fruits. Metabolite profiling was conducted using Gas Chromatography–Mass Spectrometry (GC–MS) on methanol extracts, followed by multivariate statistical analysis, including Principal Component Analysis (PCA), and a heat map visualization of compound concentrations in the two varieties. The results identified β-caryophyllene as the major secondary metabolite present in both varieties, with a relative abundance of 16.46% in the Caxcana variety and 23.06% in the S-56 cultivar. Furthermore, in silico analyses, such as network pharmacology and molecular docking, revealed key interactions with proteins such as CB2, PPARα, BAX, BCL2, and AKT1, suggesting potential therapeutic relevance. These findings highlight the pharmacological potential of guava leaf metabolites in natural product chemistry and drug discovery.

## 1. Introduction

Plants have been part of traditional medicine across various cultures over the years due to their rich content of bioactive compounds, commonly referred to as phytochemicals, primarily derived from secondary metabolism [1]. These metabolites are crucial in developing new drugs and therapeutic agents, offering individual, additive, or synergistic activities that enhance their medicinal potential [2].

The guava tree (*Psidium guajava* L.), native to South America and a member of the Myrtaceae family [3], has been particularly valued in traditional medicine. Its leaves have been extensively used in regions such as Mexico, Africa, Asia, and Central America, often prepared as tea, to address a wide range of ailments, including diarrhea, gastrointestinal disorders, dysentery, hypertension, diabetes, and dental caries, among others [4]. Rich in both macro- and micronutrients, guava leaves contain carbohydrates, carotenoids, fatty acids, polyphenols, vitamins, and minerals [5]. Furthermore, they possess a diverse array of volatile compounds [4], many of which exhibit promising therapeutic activities.

Mexico is one of the leading guava producers worldwide. This crop is characterized by significant genetic variability and diverse morphological, physical, chemical, and biological characteristics of its fruits [6]. In the “Calvillo-Cañones” region, one of the primary guava-producing regions in the country, more than 100 varieties are cultivated [7]. Among the most economically important guava varieties is the “Caxcana” cultivar, characterized by its white pulp, semi-round shape, and fruit weight ranging from 75 to 95 g [8]. This variety is distinguished by its high fruit quality, flavor, and uniformity, with over 70% classified as first-grade produce. Additionally, “Caxcana” exhibits a higher yield than other nationally cultivated guava varieties, making it a valuable option for commercial production [9].

The “S-56” accession, known for its pink pulp, produces high-quality fruits with an above-range weight of 80 to 120 g and a sweet flavor [8]. In the region, pink-fleshed guavas represent a valuable alternative for diversifying the fresh guava market and agro-industrial use [10].

Different crops and varieties of guava trees present unique metabolomic profiles in their leaves and fruits [11]. A metabolic profile provides a snapshot of the biochemical state of an individual biological sample, reflecting the genetics of each organism and the changes occurring under various physiological, environmental, and developmental conditions [12]. The metabolomic characterization of different varieties of guava tree leaves aims to identify the largest number of individual metabolites, facilitating comparison and differentiation between the varieties [13]. This process allows a detailed characterization of their phytochemical composition and the assessment of their potential activity for human health [1,2].

On the other hand, chromatographic analysis is widely recognized as one of the most important analytical chemistry tools for examining plant extracts, thanks to its simplicity, sensitivity, and effectiveness in separating components [14]. Among the various techniques available, gas chromatography coupled with a mass spectrometer (GC–MS) is particularly noteworthy for its application in studying secondary metabolites in plants [13]. This method offers excellent separation capacity, selectivity, sensitivity, and reproducibility, while also enabling the differentiation of individual plants within the same species [15].

In this study, we conducted a phytochemical characterization of two guava leaf varieties using GC–MS to identify the main components present in methanolic extracts. One of the most abundant compounds identified in both varieties was β-caryophyllene (BC). Bioinformatic analyses of pharmacological networks and molecular docking were performed with this compound to predict its potential interactions with proteins or therapeutic targets. Key metabolic pathways involving the identified bioactive compounds in guava leaves were also analyzed using databases integrating high-throughput omics data.

## 2. Results and Discussion

### 2.1. Phytochemical Screening in Guava Leaves by GC–MS Analysis

Twenty-eight chemical compounds were identified through GC–MS analysis of methanolic extracts from guava leaves (Figure 1). The identified compounds are listed in Table 1, with their respective retention time (RT), relative abundances expressed as percentage area, for both varieties, Caxcana and S-56 (%), molecular ion (M^+^), and the mass-to-charge ratios of the most abundant ions (m/z).

Three compounds were identified as primary metabolites, of which squalene was the most abundant, with a relative abundance of 10.83% in Caxcana and 13.27% in the S-56 variety. Additionally, twenty-five compounds classified as secondary metabolites were identified. Among these, β-caryophyllene was the most abundant in both varieties, with relative abundances of 16.46% and 23.06% for Caxcana and S-56, respectively. The phytochemicals exclusive to each variety are detailed in Table 1.

The relative abundance of β-caryophyllene in the guava leaf varieties analyzed in this study was higher than previously reported in GC–MS studies on guava leaves from Pakistan (15.65%) [16], Nepal (15.80%) [17], and India (15.7%) [18], but lower than those recorded in China varieties (36.9%) [19].

The significant presence of terpenes in guava leaves can be attributed to various genetic, physiological, and anatomical factors [20]. As a result, these compounds may serve as chemophenetic markers for the Caxcana and S-56 varieties [21]. In the Myrtaceae family, terpenes primarily serve as defensive agents against herbivores and pathogens [22,23]. The evolutionary diversification of these defense mechanisms drives both qualitative and quantitative variations in the phytochemical profiles of terpenes across species, varieties, populations, and even individual plants [24,25]. It has been documented that the number of terpenes present in leaves increases in response to biotic and abiotic stress factors [22,26]. For instance, compounds such as β-Himachalene and γ-Elemene, identified in extracts from various plants, have demonstrated insecticidal properties against *Spodoptera litura* larvae (Lepidoptera: Noctuidae) and the diamondback moth *Plutella xylostella* L. (Lepidoptera: Yponomeutidae) [27,28]. The high concentration of terpene compounds in the two guava leaf varieties highlights their potential as a valuable source of therapeutic agents.

### 2.2. Statistical Multivariate Analysis

Principal Component Analysis (PCA) was performed to identify key factors and visualize the relationships between chemical compounds in each biological sample [29]. PCA, a widely used tool in metabolic studies, provides a two-dimensional graphical representation (Figure 2) that reveals patterns of similarity or differentiation among the identified compounds. The analysis accounted for 96.4% of the total variance of the dataset, with the first principal component (PC1) explaining 64% of the variance and the second principal component (PC2) accounting for 32.4%. The dispersion along these axes highlights the distribution of compounds with the highest concentrations in each leaf variety. In the S-56 guava variety, β-Bisabolene (14S) had the highest concentration, with an area percentage of 9.26%, followed by (+)-Aromadendrene (8S) at 6.66%. In contrast, the Caxcana variety exhibited γ-Gurjunene (13S) as the dominant compound at 10.45% and D-Limonene (1S) at 9.76%. Both guava varieties shared β-Caryophyllene (7S) and Squalene (3P) as their most representative compounds. These findings provide valuable insights into the chemical composition and variability of the guava varieties analyzed.

The representative compounds for the S-56 variety included β-Bisabolene, which has demonstrated antimicrobial activity against *Staphylococcus aureus* [30] and cytotoxic anti-proliferative effects on breast cancer cells [31]. Additionally, (+)-Aromadendrene was identified, a compound known for its various therapeutic properties, including antimicrobial activity against *Streptococcus pyogenes* [32] and inhibition effects on the growth of skin cancer cells and HaCaT precancerous cells [33].

The representative compounds for the Caxcana variety were γ-Gurjunene and D-Limonene, a monoterpene with multiple therapeutic activities, predominantly found in plants of the *Citrus* spp. Genus. D-Limonene exhibits notable antioxidant activity, demonstrated by its protective effects against oxidative stress in lymphocyte cells induced by the exogenous addition of H_2_O_2_ [34]. This antioxidant property underscores the molecule’s potential role in several chronic degenerative disease treatments, including diabetes, cancer, chronic inflammation, and cardiovascular and gastrointestinal disorders, among others [35].

A heatmap graph was also generated to evaluate the metabolite concentrations in each sample. Figure 3 displays the combined hierarchical clustering, highlighting the relative abundance of chemical compounds across the analyzed varieties. The color scale ranges from 14, represented in intense red, to 0, depicted in dark blue, providing a clear visual representation of the variability in compound concentrations.

As highlighted in the heatmap, β-caryophyllene (BC) stands out for its high concentration in the two analyzed guava leaf varieties. This compound is also present in extracts from other plants, including *Cannabis sativa*, *Syzygium aromaticum*, *Piper nigrum*, *Rosmarinus officinalis*, *Origanum vulgare* L. [36], and species within the genus *Veronica* L. [21]. BC is a bicyclic sesquiterpene belonging to the phytocannabinoids family, recognized for its extensive range of biological activities, including anti-inflammatory, anticancer, and antioxidant properties. Furthermore, BC is used in the treatment of nervous system disorders, and atherosclerosis [37], as well as being a potent analgesic [38]. The notable presence of this compound in guava leaves further validates the traditional use of this plant for the treatment of stomach pain, underscoring its medicinal relevance.

### 2.3. Network Pharmacology and Target Protein Prediction

For this analysis, BC was selected due to its high abundance in the two guava leaf varieties. BC was used as a ligand in various databases to elucidate its pharmacological action by evaluating its binding affinity to potential human target proteins. The analysis identified several target proteins likely to interact with BC: Cannabinoid Receptor 2 (CB2, ID: 1269), Peroxisome Proliferator-Activated Receptor Alpha (PPARα, ID: 5465), Bcl2-Associated Protein X (BAX, ID: 581), B-cell Lymphoma 2 (BCL2, ID: 596), and Protein Kinase B (AKT1, ID: 207). These findings provide insights into potential mechanisms of action of BC at a molecular level.

### 2.4. Molecular Docking Analysis

The pharmacological network analysis was complemented by molecular docking studies involving BC and its target proteins. The interactions between BC and the identified proteins—CB2, PPARα, BAX, BCL2, and AKT1—are summarized in Table 2. This table provides the Vina scores, which quantify the binding affinity of BC to each target protein, alongside detailed information on the ligand-protein binding sites.

The binding prediction of BC to the CB2 receptor yielded an energy value of −8.6 kcal/mol. This interaction is characterized by polar bonds with the amino acids Thr114, Ser285, and Cys288 (Figure 4a). According to the PDBe database, 15 specific ligands have been reported for the CB2 receptor, 11 synthetic, and 4 of natural origin. The interactions observed with reported ligands, such as cannabidiol and other synthetic compounds, align with the active side interaction between Phe87 and Cys288. The activation of the CB2 receptor in humans is associated with a range of therapeutic benefits, including pain relief and the treatment of diseases linked to chronic inflammation [53]. This suggests that BC could be a promising natural remedy with both anticancer and analgesic properties [54]. Additionally, BC may offer a potential treatment for neuroinflammation related to psychiatric conditions such as depression and anxiety [53,55]. A notable example of its therapeutic potential applications is the development of BC-based ophthalmic nanoemulsions, designed to treat Acanthamoeba keratitis and relieve retinal pain in pigs [56].

The interaction between the PPARα protein and BC obtained a binding energetic value of −7.2 kcal/mol. This affinity is higher than that of other natural ligands, such as palmitic acid and arachidonic acid, but lower than resveratrol (Table 2). In the PDBe database, 43 ligands have been reported for PPARα, including 31 synthetic compounds and 12 of natural origin. Polar interactions between BC and PPARα occur at the amino acids Cys276, Gln277, Ser280, Tyr314, His440, and Tyr264 (Figure 4b), with the binding site located between residues Phe218 and Tyr 464. These results are consistent with those reported by Kamata et al., 2020 [57] who analyzed both synthetic and natural PPARα ligands. PPARα is a member of the nuclear hormone receptor superfamily and plays a crucial role in regulating oxidative stress, energy homeostasis, fatty acid metabolism, and inflammatory processes [58,59]. The anti-inflammatory effects of PPARα ligands, such as fatty acids and fibrates, have been demonstrated in various mouse models. These effects are primarily attributed to the inhibition of pro-inflammatory cytokines like IL-7 and IFNγ [60,61], as well as the reduction in leukocyte recruitment to sites of inflammation [62]. Consequently, PPARα is considered a potential therapeutic target for the treatment of neurodegenerative diseases [59], diabetes, cardiovascular diseases [63], obesity (as a hypolipidemic agent) [64], and chronic lymphocytic leukemia [65], among other conditions.

Another protein identified in this analysis as a predicted interaction partner of BC was the BAX protein. The binding energy of this interaction was calculated to be −5.5 kcal/mol, comparable to the synthetic ligand K6G and higher than those naturally occurring ligands such as dodecane and thymoquinone. According to the PDBe database, 18 ligands are associated with BAX protein including 8 synthetic ligands and 10 of natural origin. The BC binding site on BAX spans residues Pro13 to Asp159, featuring polar interactions with the amino acids Gln18, Thr22, Ser55, Gly156, and Gly157, in addition to an acidic interaction with Asp53 and Asp159 (Figure 4c). BAX, a member of the BCL2 protein family, plays a proapoptotic role and is crucial in regulating cell death [66,67]. This protein is widely expressed across various cell types, including cancer cells. In most cancer cells, such as those from lungs, colon, and breast cancer, among others; BAX is predominantly found in an inactive state [68]. Consequently, the direct activation of BAX to induce apoptosis in tumor cells presents a promising target for pharmacological cancer therapies [69].

On the other hand, the interaction between BC and the BCL2 protein displayed a binding energy of −7.1 kcal/mol, comparable to that of natural ligands but lower than synthetic ones. This suggests that BC could act as a potential antagonist of BCL2. In the PDBe database, 30 ligands are reported for BCL2, including 3 of natural origin and 27 synthetic ligands. The BC–BCL2 interaction involves the active site residues Gln60 and Met167, which aligns with the interactions observed in other reported ligands. Specifically, BC forms polar interactions with Gln60, Gly135, Tyr163, and Ser166, an acidic interaction with Asp64, and a hydrophobic interaction with Tyr163 (Figure 4d). Both BCL2 and BAX are key regulators of apoptotic pathways and belong to the family of anti-apoptotic proteins [69]. The BCL2 gene is classified as an oncogene due to its overexpression in various cancers and its association with chemotherapy resistance in tumors [70]. Inhibitory molecules targeting BCL2 have been shown to induce apoptosis in chronic lymphocytic leukemia [71] and lung cancer cell lines [72]. Pharmaceutical therapies targeting BCL2 hold promise as potential cancer treatments, either as standalone agents or in combination with chemotherapeutics to enhance their efficacy [73].

Finally, for the AKT1 protein, the binding energy interaction with BC was calculated to be −7.1 kcal/mol, a value comparable to those observed for natural ligands. In contrast, synthetic ligands exhibited binding energies ranging from −8.7 to 9.4 kcal/mol. As reported in the PDBe database, AKT1 has been associated with 40 ligands, including 9 natural and 31 synthetic ligands. Notably, these share a common binding site on AKT1, from residues Val164 to Thr291, as is the case for BC artemisinin. Detailed analysis of the BC–AKT1 interaction revealed specific polar contacts with Thr211, Tyr229, and Thr291, an acidic interaction with Glu228, and a basic interaction with Lys179 (Figure 4e). These interactions collectively stabilize the binding of BC within the AKT1 binding site. AKT1 plays a critical role in regulating cell proliferation and survival [74], serving as a key component of the PI3K/AKT signaling pathway. Dysregulation of AKT1, including abnormal activation and overexpression, has been implicated in the progression of various cancers [75], such as breast, ovarian, lung, and pancreatic cancer [76,77]. Inhibitory molecules targeting AKT1 have demonstrated potential therapeutic benefits, including the induction of apoptosis in prostate cancer cells and the inhibition of breast cancer cell growth [78,79]. These findings underscore the potential of AKT1 as a therapeutic target, particularly for interventions aimed at suppressing tumorigenesis during the early stages of cancer development [80].

### 2.5. Interaction of Bioactive Compounds 

In this analysis, we identified key proteins and genes involved in the synthesis of phytochemicals with therapeutic activity, providing valuable insights into the genetic improvement of guava tree varieties (Figure 5). Among the identified proteins, the SQS1 protein, encoded by the Squalene Synthase (SQS) gene, exhibited the highest number of interactions with bioactive compounds. This enzyme plays a vital role in the squalene and triterpenoid biosynthesis pathway [81,82]. The SQS gene is present in all plant organisms, though its expression varies significantly among species [83]. Molecular characterization of the SQS gene has been conducted in various medicinal plants to facilitate phylogenetic studies, metabolic engineering, and the production of squalene and triterpenoids in bacterial systems. Overexpression of the SQS gene has been achieved in plants such as Withania somnifera to enhance withanolide synthesis [84], in Eleutherococcus senticosus to increase phytosterol and triterpenic saponin production [85], and in hairy roots cultures of Centella asiatica to boost centelloside triterpenoid synthesis [86].

Other proteins with the highest number of interactions identified in this study were TPS23 and TPS21, members of the Terpene Synthases (TPS) gene family [87]. These proteins are precursors in the biosynthesis of thousands of terpenes found in plants, contributing to the distinctive terpene compositions observed across different taxa [88,89]. Among plant families, the Myrtaceae family is noted for having the highest concentrations and diversity of foliar terpenes. Recent studies have shown that species such as *Eucalyptus grandis*, *E. globulus*, and *Corymbia citriodora* possess the largest number of TPS genes [90]. In Brazilian guava cultivars, variations in TPS genes have been analyzed, enabling the mapping of functional genes involved in terpene biosynthesis and their use as molecular markers for genetic improvement [91]. The diverse range of terpene compounds identified in this study, along with the variability among guava varieties highlight the potential for future research on TPS genes in *Psidium guajava* L. Such investigations could deepen our understanding of metabolic profiles, elucidate their roles in adaptation mechanisms, and identify molecular targets for enhancing the production of therapeutic compounds through genetic improvement.

## 3. Materials and Methods

### 3.1. Plant Material

Young, green, firm, and spot-free leaves were collected from guava trees maintained in the germplasm bank of the National Institute of Forestry, Agricultural, and Livestock Research (INIFAP), located at the geographical coordinates 22°03′21.2″ N, 102°52′53.6″ W. Two varieties were selected for analysis: the yellow-fruited Caxcana and the pink-fruited S-56 (Figure 6). Collected leaves were washed with distilled water to remove contaminants and subsequently freeze-dried using a Labconco Freeze Dry System/Freezone 4.5.

### 3.2. Preparation of Methanol Extracts

A total of 500 mg of freeze-dried leaf material was weighed and transferred into a 50 mL Falcon tube. To each sample, 5 mL of HPLC-grade methanol (Karal, Leon, Guanajuato, Mexico) was added. The samples were sonicated at 40 °C for 20 min using a Bransonic 1800 ultrasonic cleaner (Emerson, Allen Park, MI, USA), followed by vortex mixing for 30 s. Subsequently, the samples were centrifuged at 6000 rpm for 10 min, and the supernatant was carefully collected for analysis by gas chromatography-mass spectrometry (GC–MS) using an Agilent 7890A GC system. Each extraction process was performed in triplicate for every leaf sample, and three technical replicates were analyzed for each biological replicate.

### 3.3. Analysis and Compound Identification by GC–MS Chromatography

Gas chromatography (GC) analyses were performed using an Agilent 7890A GC system coupled with a 5975C single quadrupole mass spectrometer (MS). The system was equipped with an Agilent DB-17HT capillary column (30 m × 0.25 mm × 0.15 µL), and helium served as the carrier gas at a constant flow rate of 1.29 mL/min. Samples (1 µL) were injected in splitless mode, with the injector temperature set at 250 °C. The oven temperature program was initiated at 50 °C, followed by a 10 °C /min ramp to reach 250 °C, and held for 15 min. Data acquisition and processing were conducted using Agilent MassHunter Quantitative Analysis software B.07.00, which allowed mass spectral deconvolution and integration. Compound identification was achieved by matching the acquired spectra against the NIST2011 library, considering compounds present in at least 70% of the total reads analyzed, and exhibiting a minimum similarity score of 80% for reliable identification

### 3.4. GC–MS Statistical Multivariate Analysis

Principal Component Analysis (PCA) was conducted on a data matrix comprising the percentage area of each chemical component identified in the guava leaf varieties. Data normalization and scaling were carried out using the Pareto scaling method [92]. Additionally, hierarchical clustering analysis (HCA) was performed, accompanied by a heat map to visualize the metabolite concentrations across the guava varieties. Euclidean distance was used as the metric for clustering. Data matrices for PCA and HCA were exported to the ClustVis (http://biit.cs.ut.ee/clustvis/ (accessed on 10 February 2025)), for graph generation and visualization.

### 3.5. Molecular Docking Analysis 

The 3D structures of the target proteins CB2 (ID:6PT0), PPARα (ID:6KXY), BAX (ID:6EB6), BCL2 (ID:5FCG), and AKT1 (ID:4EKL) were retrieved from Protein Data Bank (www.pdb.org). Protein preparation for molecular docking analysis was carried out using MOE 2019 software. The chemical structure of BC (ID:6PT0) was obtained from PubChem (https://pubchem.ncbi.nlm.nih.gov/ (accessed on 18 January 2025)). Docking studies were conducted using the cb-Dock2 platform (https://cadd.labshare.cn/cb-dock2/index.php/ (accessed on 18 January 2025)) following standard procedures. Interaction diagrams and 2D images depicting the ligand-target protein interactions were generated using MOE 2019 based on the docking simulation results.

### 3.6. Interaction of Bioactive Compounds 

The identified phytochemicals were analyzed in silico using the STITCH 4 database (http://stitch.embl.de/ (accessed on 18 January 2025)), which integrates protein–chemical interactions with user-provided data.

## 4. Conclusions

In this study, we identified several metabolites with therapeutic potential, mainly from the terpene family, with β-caryophyllene (BC) being the main component in the leaves of the two varieties analyzed. This study provides valuable insights into the metabolomic composition of guava leaves from the “Calvillo-Cañones” region. PCA, pharmacological network bioinformatics, and molecular docking approaches uncovered specific metabolites with potential therapeutic relevance. Notably, the predicted interactions of BC with key target proteins (CB2, PPARα, BAX, BCL2, and AKT1) suggest its involvement in crucial biochemical pathways related to inflammation, apoptosis, and cellular survival. These findings bridge an important gap in understanding guava leaf metabolites and their pharmacological potential. The identification of cultivar-specific compounds further refines our knowledge of the phytochemical diversity within *Psidium guajava* L. Additionally, our results provide a foundation for future research on the biomedical applications of guava-derived terpenes, particularly in drug discovery and therapeutic development. Further experimental validation of these interactions will be essential to fully elucidate their pharmacological significance and clinical applicability.

## Figures and Tables

**Figure 1 ijms-26-02530-f001:**
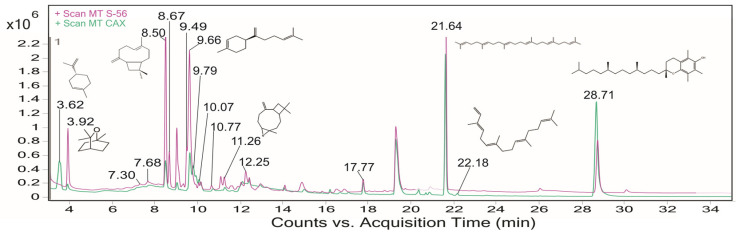
Total ion chromatogram obtained by analysis of methanolic extracts from guava leaves. The pink chromatogram corresponds to the S-56 variety, while the green represents the Caxcana cultivar.

**Figure 2 ijms-26-02530-f002:**
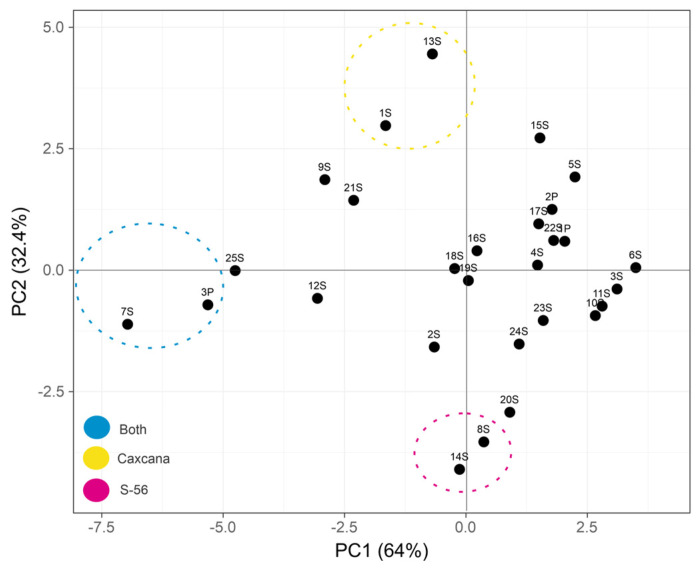
Principal Component Analysis (PCA) biplot, based on the variability of phytochemical profiles in the leaves of Caxcana and S-56 guava varieties. Created using ClustVis. (https://biit.cs.ut.ee/clustvis/, accessed on 10 February 2025).

**Figure 3 ijms-26-02530-f003:**
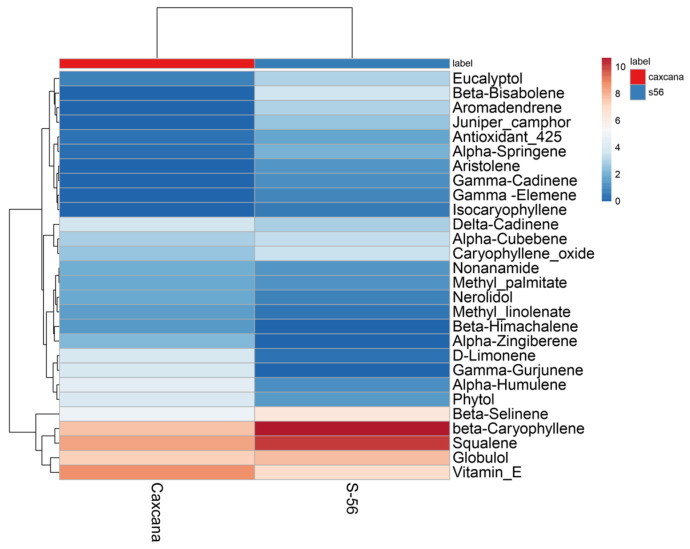
Heatmap with a color gradient from red to blue illustrates the relative abundance of chemical compounds from high to low in Caxcana and S-56 varieties of *P. guajava* L. leaves. Created using ClustVis.

**Figure 4 ijms-26-02530-f004:**
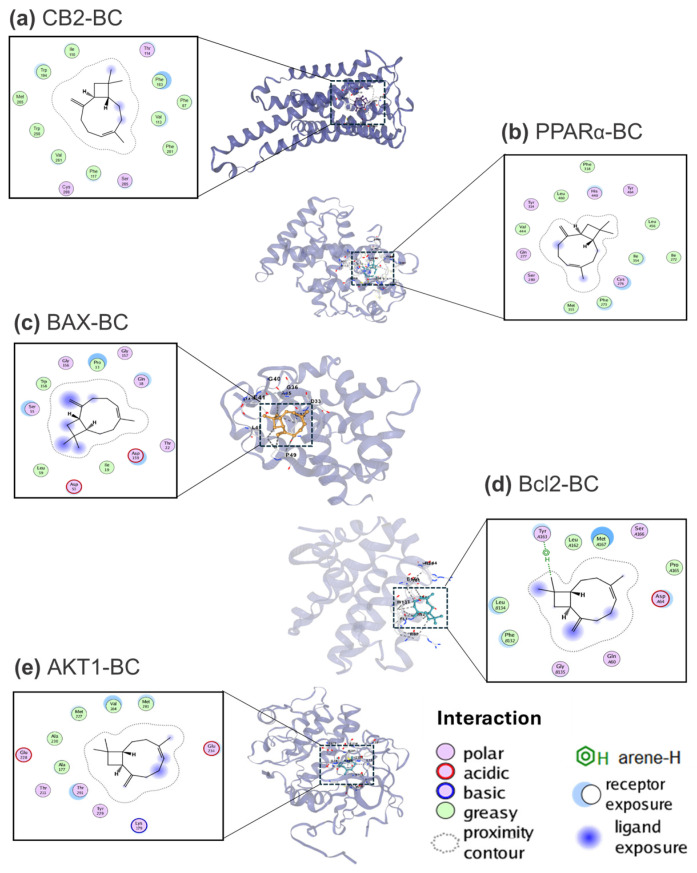
Two-dimensional and 3D models of protein–ligand active site interactions: molecular docking of BC with the following proteins: (**a**) CB2, (**b**) PPARα, (**c**) BAX, (**d**) BCL2, (**e**) AKT1.

**Figure 5 ijms-26-02530-f005:**
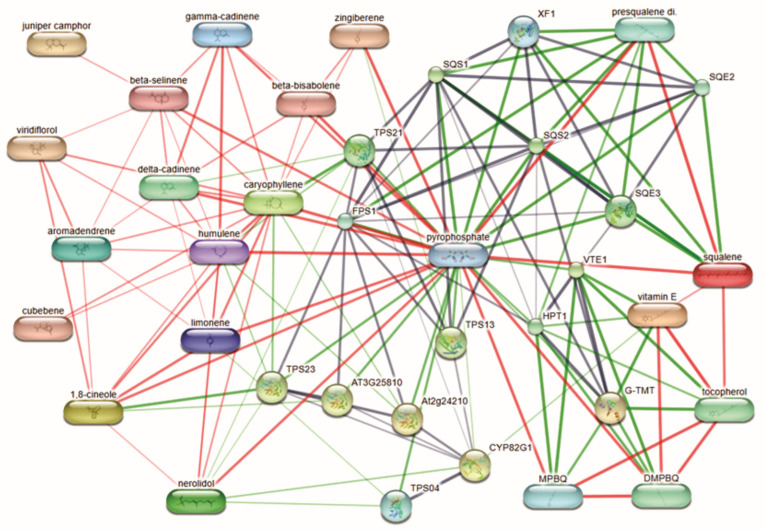
Interaction network of bioactive metabolites in *Psidium guajava*. Stronger associations are represented by thicker lines: protein-protein interactions are shown in gray, compound–protein interactions in green, and compound–compound interactions in red.

**Figure 6 ijms-26-02530-f006:**
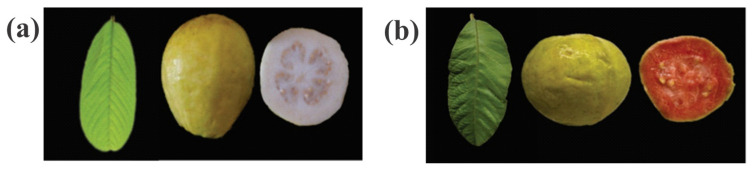
Guava fruit and leaf varieties from the “Calvillo-Cañones” region. (**a**) Caxcana cultivar. (**b**) S-56 accession.

**Table 1 ijms-26-02530-t001:** Characterization of phytochemical compounds in methanolic extracts of guava leaves varieties Caxcana and S-56.

Code	RT	CAS	Compounds	% Area Caxcana	% Area S-56	M^+^	m/z (Abundance)
Primary Metabolites							
1P	14.07	112-39-0	Methyl palmitate	0.80	0.54	270.3	74.0
2P	16.21	301-00-8	Methyl linolenate	1.32	0.36	292.2	79.1
3P	21.64	7683-64-9	Squalene	10.83	13.27	410.4	69.1
Secondary Metabolites							
1S	3.62	5989-27-5	D-Limonene	9.76	0.96	136.1	68.1
2S	3.92	470-82-6	Eucalyptol	1.28	4.65	154.1	43.0
3S	7.30	3242-08-8	**γ-Elemene**	np	0.56	204.2	121.1
4S	7.68	17699-14-8	α-Cubebene	0.92	1.07	204.2	119.1
5S	8.16	1461-03-6	**β-Himachalene**	1.4	np	204.2	119.1
6S	8.31	13877-93-5	**Isocaryophyllene**	np	0.24	93.1	93.1
7S	8.50	87-44-5	β-Caryophyllene	16.46	23.06	204.2	93.1
8S	8.67	72747-25-2	**(+)-Aromadendrene**	np	6.66	204.2	161.1
9S	9.02	6753-98-6	α-Humulene	10.28	2.67	204.2	93.1
10S	9.09	6831-16-9	**(-)-Aristolene**	np	1.17	204.2	91.1
11S	9.33	39029-41-9	**γ-Cadinene**	np	0.89	204.2	161.1
12S	9.49	17066-67-0	β-Selinene	5.20	6.85	204.2	105.1
13S	9.59	22567-17-5	**γ-Gurjunene**	10.45	np	204.2	189.2
14S	9.66	495-61-4	**β-Bisabolene**	np	9.26	204.2	93.1
15S	9.79	495-60-3	**α-Zingiberene**	3.84	np	204.2	119.1
16S	10.07	483-76-1	δ-Cadinene	1.99	1.58	204.2	161.1
17S	10.77	142-50-7	Nerolidol	1.33	0.58	222.2	69.1
18S	11.10	489-41-8	(-)-Globulol	2.09	2.21	222.2	43.1
19S	11.26	1139-30-6	Caryophyllene oxide	1.66	2.22	220.2	79.1
20S	12.25	473-04-1	**Juniper camphor**	np	4.69	222.2	43.0
21S	12.42	150-86-7	Phytol	7.50	2.59	296.3	68.1
22S	17.77	1120-07-6	Nonanamide	0.94	0.62	157.1	59.0
23S	20.70	88-24-4	Antioxidant 425	0.35	1.83	368.3	191.1
24S	22.18	77898-97-6	α-Springene	0.37	2.59	418.6	149.1
25S	28.71	59-02-9	Vitamin E	11.24	8.88	430.4	165.1

RT = retention time; M^+^ = molecular ion; np = not present; Compounds exclusive to the Caxcana variety are indicated in bold type. Compounds exclusive to the S-56 variety are noted in red.

**Table 2 ijms-26-02530-t002:** Results of molecular docking studies: binding interactions of β-Caryophyllene with key ligands of target proteins CB2, PPARα, BAX, BCL2, and AKT1.

CB2 Protein						
Compound (Ligand)	PubChem ID	Origin	Binding Affinity (kcal/mol)	Cavity Size	Aminoacidic Interaction	Reference
β-caryophyllene	5281515	Natural	−8.6	1654	87–288	
Quercetin	5280343	Natural	−8.8	1654	24–288	[39]
Codeine	5284371	Natural	−9.1	1654	110–288	[40]
Cannabidiol	644019	Natural	−9.3	1654	87–288	[41]
HU-308	5311172	Synthetic	−9.5	1654	87–288	[41]
JWH-015	4273754	Synthetic	−10.6	1654	87–285	[41]
Levonantradol	5361881	Synthetic	−11.0	1654	87–288	[40]
**PPAR** **α protein**						
β-caryophyllene	5281515	Natural	−7.2	3640	272–464	
Arachidonic Acid	444899	Natural	−7.0	3640	220–464	[42]
Resveratrol	445154	Natural	−8.4	3640	269–464	[43]
Fenofibric acid	64929	Synthetic	−7.8	3640	269–464	[44]
Bezafibrate	39042	Synthetic	−8.8	3640	272–464	[44]
Pemafibrate	11526038	Synthetic	−9.2	3640	218–355	[44]
**BAX protein**						
β-caryophyllene	5281515	Natural	−5.5	183	13–159	
Thymoquinone	10281	Natural	−5.3	148	13–159	[45]
Kaempferol	5280863	Natural	−6.3	326	13–159	[46]
Quercetin	5280343	Natural	−6.4	180	13–159	[45]
BAM7	3101542	Synthetic	−7.2	148	28–61	[47]
SMBA1	6070109	Synthetic	−7.4	148	29–61	[47]
BTSA1	3857348	Synthetic	−7.8	183	29–61	[47]
**BCL2 protein**						
β-caryophyllene	5281515	Natural	−7.1	124	60–167	
Maytansine	5281828	Natural	−7.2	272	65–114	[48]
Annocatalin	44566987	Natural	−7.3	272	69–114	[48]
Ginsenoside Rg1	441923	Natural	−8.4	272	60–167	[49]
Obatoclax	11404337	Synthetic	−7.5	94	65–114	[50]
Navitoclax	24978538	Synthetic	−9.2	272	65–114	[50]
Sonrotoclax	149553242	Synthetic	−9.4	91	65–114	[50]
**AKT1 protein**						
β-caryophyllene	5281515	Natural	−7.1	1673	164–291	
Tehranolide	6711941	Natural	−7.3	1673	156–438	[51]
Artemisinin	68827	Natural	−7.8	1673	156–292	[51]
Shogaol	5281794	Natural	−7.5	1673	156–438	[51]
XM1	46870040	Synthetic	−8.7	1668	156–442	[52]
Ipatasertib	24788740	Synthetic	−8.7	1668	156–442	[52]
Uprosertib	51042438	Synthetic	−9.4	1668	156–438	[52]

## Data Availability

Data are contained in the article.

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
