# Peer review of "Metabolomic Characterization and Bioinformatic Studies of Bioactive Compounds in Two Varieties of Psidium guajava L. Leaf by GC–MS Analysis"

_ijms, 2025, doi:10.3390/ijms26062530_

Round 1
Reviewer 1 Report
Comments and Suggestions for Authors
Dear Ana,
Thank you for submitting your manuscript, Metabolomic Characterization and Bioinformatic Studies of Bioactive Compounds in Two Varieties of Psidium guajava L. Leaf by GC-MS Analysis, to International Journal of Molecular Science. I appreciate the opportunity to review your work.
Overall, I found your manuscript to be interesting but requiring some improvements.
Here are my specific comments and suggestions:
Abstract section
The author should write the abstract in this format: A brief introduction followed by the aim of the study, the methodology used, the key results from the study, and conclusion.
Line 19-28 : The authors should rewrite this
For instance
This analysis was conducted using Gas chromatography coupled with Mass Spectrometry (GC-MS) should be followed by the statement a multivariate statistical analysis using Principal Component Analysis (PCA). Additionally, a hierarchical analysis combined with a heat map was used to visually indicate the concentration values of the compounds in the two varieties. Furthermore, in silico studies such as network pharmacology and molecular docking studies were also performed to highlight the potential of these guava leaf varieties in natural product chemistry and pharmacology. The result showed that β-caryophyllene was the major secondary metabolite present, with a relative abundance of 16.46% in the Caxcana variety and 23.06% in the S-56 variety. Write the key result obtained in the PCA analysis and the heat map analysis. Also write the key result from the molecular docking study. Finally, write a brief conclusion.
Line 94-96: The orange chromatogram is not clear in Figure 4. I suggest the authors should use another color that can contrast well with pink.
Line 179-180 (Table 2). Are the other compounds in the table, apart from β-caryophyllene, gotten from the work of previous authors, or was the study conducted in this work? If the compounds were retrieved from past works, please the author should cite appropriate references.
Line 184: (figure 4: A) should be changed to ( figure 4A )
Line 222: (figure 4:C) should be changed to ( figure 4C )
Line 236: (figure 4: D) should be changed to ( figure 4D )
Line 251: The authors should space site.AK71.
I hope you find my comments helpful in revising your manuscript.
Author Response
"Please see the attachment"

Reviewer 2 Report
Comments and Suggestions for Authors
The manuscript is generally well written and deals with an interesting topic. However, before it can be considered for review, certain aspects must be reviewed.
Introduction
In the introduction if varieties and related characteristics of Psidium guajava L. are fully described, a review of the state of the art concerning the characterization of these plants is lacking.
In addition, in the introduction it must be evident the objective of the study and the advantages it may bring over the current literature. While the description of the individual analyses done is either better integrated within the discourse or does not need to be reported
Line 39: please change “its” with “their”
Line 45: add the dot at the end of the sentence
Line 46: Please add the reference
Result and discussion
In general, the results are well presented but the discussion is, in some places, lacking by not referring to other studies in the literature that have performed the same analyses although on different matrices
Lines 89 – 92: delete the following lines from the main text. the description of the colors and style used in writing the table should be entered as a footnote to the table itself
Table 1: for those m/z values that have “0” as decimals please add it
Materials and methods
Line 296: Please specify the parameters for those you define a leaf “healthy”
Line 301: please specify the company
Line 316: please specify the company
Line 317: please specify the company and the temperature used for the column
Line 322: please specify the company of the Analysis software
Line 329: please add the reference for the “Pareto scaling method”
Conclusion
In the conclusion add the advantages that the results obtained have brought in bridging the current scientific knowledge in this area
Author Response
"Please see the attachment"

Round 2
Reviewer 2 Report
Comments and Suggestions for Authors
The authors have responded fully to the comments made and have made appropriate changes to the manuscript making it complete and acceptable in the present form for publication